# Virome Profiling of an Eastern Roe Deer Reveals Spillover of Viruses from Domestic Animals to Wildlife

**DOI:** 10.3390/pathogens12020156

**Published:** 2023-01-18

**Authors:** Yue Sun, Lanshun Sun, Sheng Sun, Zhongzhong Tu, Yang Liu, Le Yi, Changchun Tu, Biao He

**Affiliations:** 1Changchun Veterinary Research Institute, Chinese Academy of Agricultural Sciences, Changchun 130122, China; 2Provincial Wildlife Disease Monitoring Station of Shuanghe, Xunke 164400, China; 3Jiangsu Co-innovation Center for Prevention and Control of Important Animal Infectious Diseases and Zoonosis, Yangzhou University, Yangzhou 225009, China

**Keywords:** Eastern roe deer, virome, parvovirus, kobuvirus, spillover

## Abstract

Eastern roe deer (*Capreolus pygargus*) is a small ruminant and is widespread across China. This creature plays an important role in our ecological system. Although a few studies have been conducted to investigate pathogens harbored by this species, our knowledge of the virus diversity is still very sparse. In this study, we conducted the whole virome profiling of a rescue-failed roe deer, which revealed a kobuvirus (KoV), a bocaparvovirus (BoV), and multiple circular single-stranded viruses. These viruses were mainly recovered from the rectum, but PCR detection showed systematic infection of the KoV. Particularly, the KoV and BoV exhibited closely genetic relationships with bovine and canine viruses, respectively, highly suggesting the spillover of viruses from domestic animals to wildlife. Although these viruses were unlikely to have been responsible for the death of the animal, they provide additional data to understand the virus spectrum harbored by roe deer. The transmission of viruses between domestic animals and wildlife highlights the need for extensive investigation of wildlife viruses.

## 1. Introduction

China is one of the countries with the most diverse and abundant wildlife resources, in which dwells over 7300 vertebrate species, comprising ~11% of the world’s total wildlife species [1]. Owing to a series of national regulations and laws to ban illegal hunting and overexploitation, and the commitments of the Chinese government and scientific community to biodiversity conservation, the living status of many wild animals has been greatly improved [1]. However, there are still many other indigenous species at risk of extinction, for example, a piece of unfortunate information from the most recent International Union for Conservation of Nature and Natural Resources (IUCN) report has formally announced the extinction of a specific paddlefish (*Psephurus gladius*) of the Yangtze river [2]. Therefore, the wildlife conservation has a long way to go in China.

Among the challenges faced by wildlife conservation, pathogens constitute an overlooked but substantial portion. Indeed, they pose a great risk to wildlife, for example, influenza viruses in birds [3], African swine fever viruses in wild boars [4]. Wildlife-borne pathogens also severely threaten global public health. Particularly, spillovers of some deadly wildlife-borne viruses, such as the Nipah virus, Marburg virus, and Rabies virus, have caused several outbreaks of emerging/re-emerging infectious diseases (EIDs) in humans [5]. Thus, investigation of wildlife viruses is not only an important measure in wildlife conservation, but also helps to control and prevent the potential outbreaks of wildlife-originated zoonosis [6].

Belonging to the genus *Capreoplus* within the family Cervidae, Eastern roe deer (*Capreolus pygargus*) are one of the most widespread and abundant free-living ungulates in China, and its population status serves as a biological indicator of the environmental health [7]. They inhabit different types of deciduous and mixed forests and forest-steppes, and are apt to frequently contact with domestic animals, providing ample opportunities for pathogens to transmit between roe deer and other animals. Recent investigation has shown that infectious diseases are an important factor causing the death of roe deer, including parasitic infections and bacterial diseases [8]. Besides, a few serological and molecular investigations have revealed that the creatures are also infected by the tick-borne encephalitis virus, hepatitis E virus, Schmallenberg virus, and so forth, indicating that roe deer are also involved in the circulation of these zoonotic viruses [9]. However, all those investigations used pathogen-specific approaches and are fragmental, and apparently, the complete spectrum of pathogens harbored by this species remains largely undetermined. In this study, we examined the complete virome of a dead roe deer using a DNA-specific multiple displacement amplification (MDA) and an RNA-specific meta-transcriptomic (MTT) method. Results provide additional knowledge of the genetic diversity of roe deer viruses.

## 2. Materials and Methods

### 2.1. Sample Collection

In February 2021, an injured female adult roe deer was found in the field in Xunke county, Heilongjiang province, and was transported to the Provincial Wildlife Disease Monitoring Station of Shuanghe for rescue. The animal was very thin and suffering severe injuries in its hip and died very soon after its arrival. A necropsy was immediately performed, which revealed no abnormal lesions in its internal organs but a lack of food in its gastroenteric track was noted. Its rectum, cervical lymph nodes, kidney, lung, brain, and liver were sampled and cryo-transported to the laboratory for a viromic analysis. The species was morphologically identified by the staff at the station and further confirmed by sequencing the mitochondrial cytochrome coxidase subunit I gene (COI) [10].

### 2.2. Sample Pretreatment and High-Throughput Sequencing

A small piece (~0.2 g) of each tissue was cut and homogenized with sterile PBS. After centrifugation, supernatants were passed through 0.45-μm-pore-size membranes (Millipore, Boston, MA, USA), and digested with nuclease to eliminate the contamination of foreign nucleic acids. For MTT sequencing, RNA was extracted using Trizol reagent (Invitrogen, Carlsbad, CA, USA), and subjected to rRNA depletion using an Ribo-ZeroTM Magnetic Gold Kit (Epicentra Biotechnologies, Madison, WI, USA), followed by RNA-sequencing on an Illumina NovaSeq sequencer using an NEBNext ultra-directional RNA library prep kit (NEB, Ipswich, MA, USA). However, for MDA processing, DNA was extracted using a DNeasy Blood and Tissue kit (Qiagen, Hilden, Germany) and amplified using an illustra GenomiPhi V2 DNA amplification kit (GE, Fairfield, CT, USA) as per the manufacturer’s manual. The products were purified using a QIAquick PCR Purification Kit (Qiagen, Hilden, Germany) with one μg used to Illumina pair-end (150 bp) sequencing at an Illunima NovaSeq 6000 sequencer. To inspect any possibilities of cross-contamination during sample pretreatment and sequencing, samples of an Amur leopard cat were simultaneously processed [11].

### 2.3. Virome Annotation

The raw data were processed using fastp version 0.19.7, and subjected to host sequence removal by mapping against the whole genomic assembly of *Capreolus pygargus* (accession number: GCA_012922965.1) using bowtie2 version 2.4.1, followed by a rapid metagenomic classification of bacterial, archaeal, and fungal genomes using kraken2 version 2.0.9. The remaining reads of RNA and DNA viromes were respectively mixed together and de novo assembled using metaSPAdes version 3.14.9. These contigs were annotated using blastn and diamond blastx searching (e-value ≤ 1 × 10^−10^) against our refined eukaryotic viral reference database (EVRD)-nt/aa version 1.0 [12]. To examine the authenticity of these virus-like contigs (VLCs), reads were mapped back to VLCs, and the vertical and horizontal coverages were determined using samtools version 1.10.

### 2.4. PCR Validation and Gap-Filling

To validate the viromic results and fill the genome gaps, primers (Appendix A) were designed using Primer Premier5 targeting the contigs of bocaparvovirus and kobuvirus. Viral RNA was extracted by a RNeasy Mini Kit (Qiagen, Hilden, Germany) and reverse transcription was affected with a first cDNA synthesis kit (TaKaRa, Dalian, China) according to the manufacturer’s protocol. DNA extraction is described above. Double-distilled water was used as a negative control. PCR amplification was conducted by a 2× Rapid Taq Master Mix (Vazyme, Nanjing, China) with the following program: 95 °C for 5 min, 40 cycles of denaturation at 95 °C for 15 s, annealing at 59 °C for 15 s (or adjusted according to different primer pairs) and extension at 72 °C for 25 s, and a final extension at 72 °C for 5 min. The expected products were directly sequenced on an ABI 3730 Sanger sequencer (Comatebio, Changchun, China).

### 2.5. Genomic Characterization and Phylogenetic Analyses

Open reading frames (ORFs) of viral genomes were predicted using Geneious version 4.8.3. The genomic structure was illustrated using Seqbuilder version 7.1.0. The stem-loop of each genome was predicted by mFold. Alignments of nucleotide (nt) and amino acid (aa) sequences with other representatives of known viruses were conducted using MAFFT version 7.471. TrimAI version 1.2 was used to clipping ambiguous alignments and a model finder was used to predict the best model. Phylogenies were inferred by IQ-TREE version 1.6.8 with 1000 bootstrap replicates.

## 3. Results

### 3.1. Overview of the Virome

The dead roe deer provided a rare opportunity to investigate the virus diversity harbored by this creature. Therefore, a combination of MDA and MTT technologies was employed to profile the whole eukaryotic virome, which eventually generated 37.8 (for DNA) and 28.8 (for RNA) gigabase (GB) reads with 6.3 ± 2.7 and 4.8 ± 1.0 GB per DNA and RNA library, respectively. The viromic annotation revealed 81 potentially eukaryotic VLCs with 306–6272 nt in lengths covering *Parvoviridae*, *Circoviridae*, *Smacoviridae*, *Genomoviridae*, and *Picornaviridae* (Figure 1), among which 16 were complete genomes corresponding to 14 circular single stranded DNA (cssDNA) viruses, one bocaparvovirus, and one kobuvirus. Comparison with the virome of an Amur leopard cat did not find any contigs related to the viruses identified from Amur leopard cat [11], indicating no cross-contaminations as well as contamination from reagents in sample processing. Mapping reads back to these contigs showed that the rectum sample had the most abundant viruses with ~10^6^ reads related to the genus *Gemykibivirus* within the family *Genomoviridae*. On the contrary, these solid organs were very low in richness and abundance of virus species (Figure 1). It is reported that bocaparvovirus (BoV) and kobuvirus (KoV) were associated with certain diseases in various animal species [13,14], hence we validated their presence in these samples by using PCR/RT-PCR detections. The detection results were largely consistent with the viromic analysis, that is, the samples that had reads for a virus were also positive in the specific PCR/RT-PCR detection. However, PCR/RT-PCR revealed more positives. In particular, BoV was detected in almost all sampled tissues except the liver. This discrepancy should be ascribed to how viromic analysis at such sequencing depth is less sensitive than the conventional PCR method, which can be improved by ultra-deep sequencing [15].

### 3.2. Genomic and Phylogenetic Characterization of Kobuvirus

Kobuvirus is a small, spherical, and non-enveloped picornavirus with a single-stranded positive-sense RNA as its genome. The infection of KoV is common among humans, rodents, pigs, carnivores, and ruminants. Among the six species (*Aichivirus A–F*) of the genus *Kobuvirus*, members of the species *Achivirus A* can cause acute gastroenteritis in humans and the bovine kobuvirus from the species *Achivirus B* might lead to diarrhea in cattle [16]. The MTT data generated two contigs, which were further joined by a gap-filling RT-PCR, eventually resulting in an 8299 nt-long sequence that covers the entire ORF. Online blastn search of the sequence against Genbank showed that it was 93.3% nt identical with a bovine KoV identified from a diarrheal calf in Hebei province [17]. Phylogenetic analysis based on the entire ORF nt sequences revealed that the evolution of KoVs is highly related to their hosts. Although detected in different countries and even continents, KoVs of bovine, caprine and porcine respectively fell into three independent clades, while CpKoV/XK/CHN/2021 identified in this study is closely clustered with those bovine KoVs within the species *Aichivirus B* (Figure 2a).

### 3.3. Genomic and Phylogenetic Characterization of Bocaparvovirus

BoVs, belonging to the family *Parvoviridae*, are a group of single-stranded DNA viruses. They have a wide range of hosts, including humans, cats, dogs, pigs, sheep, and cow, and can cause respiratory and gastrointestinal tract diseases in juvenile animals and humans [18]. By de novo assembly and gap-filling PCR, we obtained a nearly full-length (4923 nt) CpBoV/XK/CHN/2021 genome. The sequence comparison and phylogenetic analysis of the NS1 gene both showed that it was tightly clustered with a group of canine BoVs with a nt similarity as high as 97.9% with strains 14Q209 and 17CC0312 within the species *Carnivore bocaparvovirus 2*, where the former was detected from a dead Korean dog with unknown causation in 2014 [19], while the latter is a canine BoV but was recovered from a cat in Northeast China in 2017 [20] (Figure 2b).

### 3.4. Genomic and Phylogenetic Characterization of CRESS DNA Viruses

The cssDNA viruses revealed here are a group of circular Rep-encoding single-stranded (CRESS) DNA viruses, which are ubiquitously distributed with a variety of sources, such as plants, bird feces, and animal tissues [21]. We obtained 14 complete CRESS DNA virus genomes based on their overlapping ends, with 1512–2707 nt in lengths. Almost all of them encoded two ORFs, corresponding to replication (Rep) and capsid (Cap) proteins, with the majority arranged bidirectionally, but one, that is, CRESS/CpXKC1, had two genes arranged unidirectionally (Figure 3a). Of note is that the genome of GmV/CpXKC1 has four ORFs, with three of them encoding proteins associated to replication. Especially, the canonical Rep is split into two small portions. The odd genomic structure was confirmed by PCR validation. We also found stem-loop structures of CRESS DNA viruses between Rep and Cap ORFs, which initiate the rolling-circle replication of these genomes (Figure 3a). These genomes are very divergent from each other with 64–99% aa identities in Rep. However, the online blastn searches revealed various genetic distances to their known reference sequences. For example, five genomes, that is, GmV/CpXKC1-5, showed very close relationships (95–99% in nt) with some genomoviruses, but the remaining genomes were distantly related to those unclassified CRESS DNA viruses with 29.0–88.8% aa identities. To infer their phylogeny, a total of 143 Rep protein sequences were aligned with the counterparts of the 14 genomes, followed by a maximum likelihood phylogenetic analysis. These sequences from the families *Geminiviridae*, *Genomoviridae*, *Smacoviridae*, and *Circoviridae* are well segregated from each other to form independent phyloclades, indicating a robust phylogenetic result (Figure 3b). The 14 sequences were dispersed into different clades with above mentioned five ones clustered closely with genomoviruses with origins of plant, bird feces, and silkworms (Figure 3b). Whereas the rest cannot be assigned to any approved families but are genetically related to those unclassified CRESS DNA viruses from fish, giant panda feces, and spiders (Figure 3b).

## 4. Discussion

Here we report the viromic profiling of a rescue-failed roe deer followed by PCR/RT-PCR validation. In order to obtain the complete spectrum of viruses harbored by this animal, we employed a combination of MDA and MTT methods, which has been proven be robust to capture the whole virome [15]. Although we generated more than 60 Gb of data for the animal, very limited known eukaryotic viruses were uncovered, which should undoubtedly be attributed to the small sample size and the inability of the viromic annotation method used here to identify those remote viruses. Considering the important role of roe deer in the ecological system [7] and the paucity of our understanding of their virus diversity, viromic investigation of this species with more individuals in a broader area should be intensified in the future. Although the virome and PCR/RT-PCR detection revealed KoV and BoV in the rectum sample, both of which are potential causative agents for gastroenteritis or diarrhea, its internal organs looked very normal, with no sign of diarrhea at sampling except for the injures in its hip. Therefore, we cannot arbitrarily ascribe the death of the roe deer to any pathogen infections, but rather, it is reasonable to conclude that the injury and malnutrition led to its death. In addition, these viruses were largely limited to the intestine and the lymph node, but we noted the wide distribution of gemykibivirus, especially at high loads in the intestine and the lymph node (Figure 1). However, some tissues, like the brain, are recognized as virus-free in a healthy condition. We set a control to inspect the possible contamination events during sample pretreatment for metagenomic sequencing and found no cross-contamination happened, but we cannot rule out the possible contamination between organs by blood during necropsy.

Interestingly, the CRESS DNA viruses revealed here were very genetically diverse and showed as high as 99% nt identities to their genetic neighbors of plant-, animal feces- and silkworm-origin. Recently, the diversity of CRESS DNA viruses has been greatly expanded, partially due to the wide application of high-throughput sequencing-based virome studies [22]. They are very intricate because of their ubiquitous distribution and broad association to environmental samples, plants, and animal feces [23]. Thus, the diversity of CRESS DNA viruses should be largely explained by food and environmental factors rather than their genuine infection.

Of particular note is that we revealed the sign of cross-transmission of KoV and BoV among domestic animals and roe deer. The phylogeny of KoVs is partially shaped by their hosts (Figure 2a), but the roe deer KoV fell into the clade of bovine KoVs with as high as 93.3% nt identity to a virus of diarrheal calf (Figure 2a), suggesting a possible spillover of bovine KoV to roe deer. This speculation was further verified by the BoV, which was phylogenetically surrounded by canine viruses and showed ≥93% nt identities to two viruses (Figure 2b). Especially, one of its genetic neighbors, that is, 17CC0312, was also detected in northeast China [19]. Accordingly, it is highly suggested that there be a spillover event of canine BoVs to roe deer. Such a phenomenon was also observed in our previous viromic examination of an Amur leopard cat, in which the anelloviruses and bocaparvovirus were possibly transmitted from other feline species including domestic cats [11].

Taken together, these findings conclusively show the frequent cross-transmissions of viruses from domestic animals to wildlife, which poses a substantial obstacle in wildlife conservation as well as in disease control and prevention in domestic animals. On one hand, some viruses are lethal to wildlife and would cause catastrophic disasters to wildlife. For example, the spillover of African swine fever viruses from domestic pigs to wild boars has killed a great number of wild boars [4]. On the other hand, the cross-transmission would result in the circulation of some viruses in wildlife, which carries the potential risk of reintroduction of these viruses into domestic animals, making the eradication of related diseases very difficult [24]. To tackle these issues, it is critical to stop the transmission route of viruses between domestic animals and wildlife. For such purposes, it is feasible to set up wildlife reserves and popularize house feeding of domestic animals. Besides, active vaccination of wildlife can be considered for some cases. For example, vaccination of carnivores against rabies virus have been proven to be an effective way to eliminate rabies [25]. In addition, we should intensify the investigation of wildlife viruses to improve our current fragmented knowledge about them. The unbiased virome techniques would be a wise choice in such a campaign.

## Figures and Tables

**Figure 1 pathogens-12-00156-f001:**
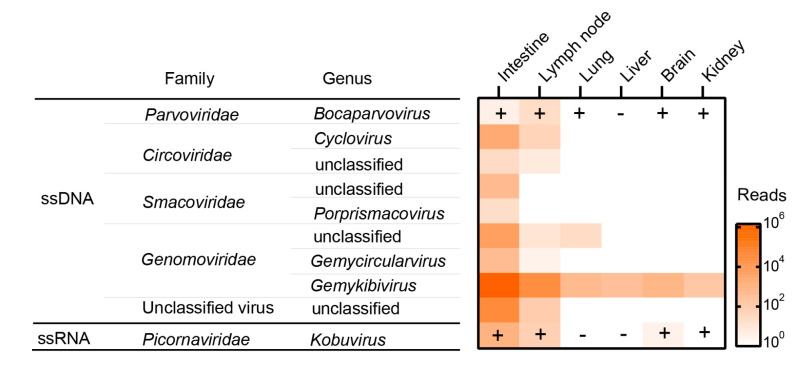
Overview of the virome and PCR confirmation of the roe deer. ssDNA: single-stranded DNA viruses; ssRNA: linear single-stranded RNA viruses; the plus and minus symbols indicate the results of PCR detection.

**Figure 2 pathogens-12-00156-f002:**
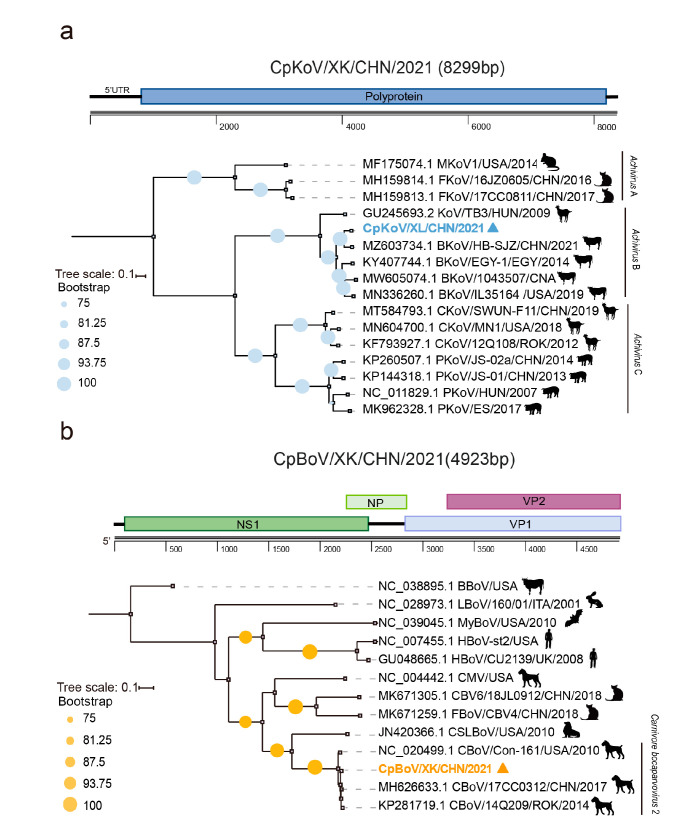
Genomic and phylogenetic characterization of the kobuvirus (in blue) (**a**) and the bocaparvovirus (in orange) (**b**) contigs. Bootstrap values of ≥75 are shown next to nodes. Trees are rooted at midpoint.

**Figure 3 pathogens-12-00156-f003:**
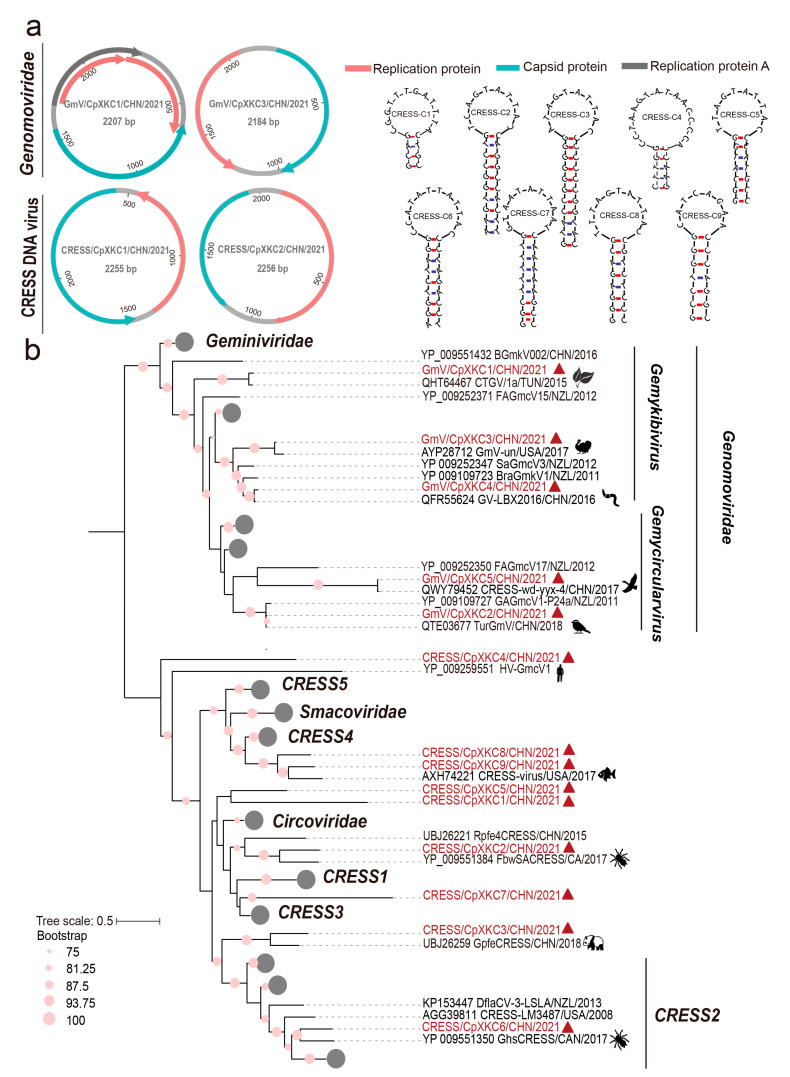
Genomic and phylogenetic characterization of diverse CRESS DNA viruses. (**a**) The predicted schematic representations of four CRESS DNA virus genomic representatives and the stem loop structures of CRESS/CpXKC1-9. Among the 14 CRESS DNA viruses, GmV/CpXKC2-5 had identical genomic structures, which are illustrated using that of GmV/CpXKC2. Similarly, we used CRESS/CpXKC1 to represent the genome structures of CRESS/CpXKC1 and CRESS/CpXKC3; CRESS/CpXKC2 to represent the genome structure of CRESS/CpXKC4-9. (**b**) Phylogenetic analysis of 14 CRESS DNA viruses based on Rep protein sequences. Sequences that are well segregated into the families *Geminiviridae*, *Smacoviridae*, and *Circoviridae* are collapsed. Sequences identified in this study are highlighted in vermilion, with the origins of their genetic neighbors indicated by animations. Bootstrap values of ≥75 are shown next to nodes. The tree is rooted at midpoint.

## Data Availability

The complete genomes of the Gmv/CpXKC1-5 and CRESS/CpXKC1-9 were deposited in Genbank under accession numbers ON554189-ON554202. The nearly complete genomes of the CpKoV/XK/CHN/2021 and CpBoV/XK/CHN/2021 were deposited in Genbank under accession numbers OP920760 and OP897810. The MDA and MTT raw data were available in the NCBI Sequence Read Archive (SRA) under accession numbers PRJNA842839.

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
