# Peer review of "Virome Profiling of an Eastern Roe Deer Reveals Spillover of Viruses from Domestic Animals to Wildlife"

_pathogens, 2023, doi:10.3390/pathogens12020156_

Round 1

Reviewer 1 Report

The article is well written and presents interesting results, however nothing as innovative or different in the literature. It is, however, a well-written article with a well-applied scientific method.

I did not find table 1 in the pdf file sent. Perhaps there is a lack and it is essential that the primers are demonstrated.

Author Response

Response (R) 1: We are sorry for the lack. We have provided Supplementary Table.

Reviewer 2 Report

The manuscript by Sun et al. describes the results of a virome investigation performed on tissue samples collected from a roe deer that died because of physical injuries and starvation. The authors found several viruses, mostly circular single stranded DNA viruses, and – as expected – most of the hits were found in the intestine. For some viruses for which the complete genome was obtained, a full genome characterization and phylogenetic analysis was performed. Additionally, the authors evaluated by PCR the tissue distribution of two animal viruses, canine bocavirus and bovine kobuvirus, confirming active infection and concluding possible cross-species transmission. The study is well performed, methods are appropriate, and conclusions seem solid. The paper is well written, and it reads nicely, although there are a few weird sentences that should be rephrased. Finally, there are a few points that I think deserve some extra care and I would like to suggest some additional discussion points. Specific comments:

- Figure 3. CRESS-viruses are characterized by the presence of a DNA sequence that folds into a stem-loop structure. It would be good if you could identify this in all your viruses and show it in the figure, to make sure that all genomes are actually CRESS viruses and not viruses with non-circular genomes that have been accidentally circularized because of redundant termini. This is especially true for CRESS/CpXKC4/CHN/2021 as it is very divergent from everything else. I would also include the genomic structure of all identified viruses or at least indicate in the figure which viruses have similar structure.  

- There is a discrepancy between Figure 1 and 3. In figure 1 you show that you identified also circoviruses and smacoviruses, but they are not depicted in Figure 3. Is that because you did not obtain the full genome? If so, this has to be specified. In any case, I would also include incompletely sequenced viruses (e.g., others for which the complete genome is not available, but you could sequence the Rep gene).

- I agree with your conclusion that most of the identified viruses are probably diet-related as they are mostly present in the intestine. However, in Figure 1, you clearly show that reads matching to Gemykibivirus could be identified in all tissues and at pretty high lads. This was not even observed for boca and kobu! I would seriously consider investigating this virus/these viruses more thoroughly because they could actually be deer viruses (unless they are reagent contaminants, but this can be excluded by looking at the leopard sequences). This could be a quite intriguing result… were all the reads from one single virus? Where was this virus/these viruses phylogenetically located (is it shown in Figure 3?)? Which viruses are the closest relatives? I also suggest discussing this hypothesis in the discussion.

- Line 218. This sentence is not clear and it should be rephrased. Also, The conclusion that the animal was healthy just because no viruses were identified is wrong. It is possible that viruses were present but were not detected. For example, very divergent viruses could have been missed or viruses at a lower load could have gone undetected because of the low sensitivity of the detection methods. This should be highlighted in the discussion.

- Although it is really reasonable that most of the viruses you found were limited to the intestine and the lymph node, have you considered the hypothesis that some tissues could have been virus-free and were accidentally contaminated during necropsy? If you think that this is not the case because precautions were taken during sample collection, it may be worth highlighting it.

Minor:

- Line 50. Cervidae should not be written in italics since for eukaryotes only species and genus names are italicized.

- Line 133. It has not been confirmed that eukaryotes are the host for Smacoviruses; I would add the word “potentially” before “eukaryotic viruses”.

- Line 137. This is also important because it excludes that viral reads are derived from reagents used during sample processing (e.g., DNA isolation columns).

- Figure 1. I would include additional labelling on the trees that specify the virus species and indicate the hosts more clearly as some of the figurines are not entirely clear.

- Trees. Were the trees rooted at midpoint (since there is no outgroup)? Maybe this can be specified.  

- Sections 3.2 and 3.3. I would clearly indicate the viral species to which the identified viruses belong.

- Lines 186-7. Maybe you could add these primers to supplementary material too.

- Lines 190 and 274-5. The ~ should be substituted by a -.

- Figure 3. It would be handy if you could indicate the different Genomoviridae genera on the tree.  

Reviewer 3 Report

Please reframe some sentences as marked on the manuscript for easy understanding of the reader.

Author Response

Thank you for your advice. We have revised the manuscript.

Round 2

Reviewer 2 Report

I am satisfied with the revisions, except for one minor thing: if the divergent CRESS viruses did not cluster within the Circoviridae and Smacoviridae it is not entirely fair to label them as Circoviridae and Smacoviridae in Figure 1….

Author Response

I am satisfied with the revisions, except for one minor thing: if the divergent CRESS viruses did not cluster within the Circoviridae and Smacoviridae it is not entirely fair to label them as Circoviridae and Smacoviridae in Figure 1….

R: Although those full-length CRESS DNA viruses did not cluster with known members of Circoviridae and Smacoviridae, they were phylogenetically related to unclassified circoviruses and smacoviruses, so it is reasonable to classify them as such. Additionally, the sequences annotated to Circoviridae and Smacoviridae also include some short fragments, which had high similarity to the known Circovirus and Smacovirus. However, due to their short length, they are not the focus of our analysis.